# Intra-Articular Ultrasonography Probe for Minimally Invasive Upper Extremity Arthroscopic Surgery: A Phantom Study

**DOI:** 10.3390/jcm12175727

**Published:** 2023-09-02

**Authors:** Shintaro Oyama, Nobuo Niimi, Masato Mori, Hitoshi Hirata

**Affiliations:** 1Innovative Research Center for Preventive Medical Engineering, Institute of Innovation for Future Society, Nagoya University, Tokai National Higher Education and Research System, NIC#5, Furo-cho, Chikusa-ku, Nagoya 4648601, Aichi, Japan; 2Graduate School of Medicine, Nagoya University, 65 Tsurumai-cho, Showa-ku, Nagoya 4668550, Aichi, Japan; h-hirata@med.nagoya-u.ac.jp; 3Planning and Product Development Department, Nippon Sigmax Co., Ltd., 1-24-1 Nishi-Shinjyuku, Shinjyuku-ku, Tokyo 1600023, Japan

**Keywords:** arthroscopic ultrasonography, medical image processing, intra-articular examination probe, phantom study

## Abstract

Background: Upper extremity arthroscopic surgery is a highly technique-dependent procedure that requires the surgeon to assess difficult cartilage conditions and manage the risk of iatrogenic damage to nerves and vessels adjacent to the joint capsule in a confined joint space, and a device that can safely assist in this procedure has been in demand. Methods: In this study, we developed a small intra-articular ultrasound (AUS) probe for upper extremity joint surgery, evaluated its safety using underwater sound field measurement, and tested its visualization with a phantom in which nerves and blood vessels were embedded. Results: Sound field measurement experiments confirmed the biological safety of the AUS probe’s output, while confirming that sufficient output power level performance was obtained as an ultrasound measurement probe. In addition, images of blood vessels and nerves were reconstructed discriminatively using A-mode imaging of the agar phantom. Conclusions: This study provides proof-of-concept of the AUS probe in upper extremity surgery. Further studies are needed to obtain approval for use in future medical devices.

## 1. Introduction

Arthroscopic surgery is an essential, minimally invasive surgical procedure for orthopedic diseases such as meniscus injuries, ligament injuries, osteoarthritis, rheumatoid arthritis, synovitis, free bone fragments, cartilage lesions, and bone tumors. It is also used as an examination technique for accurate assessment of the condition of intra-articular tissues such as cartilage, synovial membrane, and joint ligaments [1]. Recently, the demand for arthroscopic surgery has increased owing to an increase in joint diseases associated with the aging population and the general public’s growing awareness of arthroscopic surgery. In particular, upper extremity arthroscopic surgeries are increasingly indicated to maintain high upper limb function not only in athletes and active populations but also in elderly patients, and the indications have expanded to address complex conditions such as free cartilage resection, synovectomy, ligament and articular repairs [2], and intra articular fracture fixations [3]. However, due to the tight space between the joints of the upper limb and the complexity of the shape and adjacent musculotendinous and neurovascular structures, the technical requirements for upper extremity arthroscopic surgeries are high, and not many surgeons are yet trained enough to perform the surgery yet [4,5]. In addition, the field of view may be narrow especially in stiff joints, making it difficult to adequately evaluate intra-articular structures, including cartilage and joint capsules, and they are prone to procedural errors such as excessive resection and intra-articular bleeding, which may result in damage to important extra-articular structures [6]. Complications such as contractures owing to iatrogenic damage to the joint capsule, ligaments, and cartilage; postoperative ganglions; damages to nerves and vessels have also become an issue [4,7]. In particular, postoperative contractures due to excessive tissue resection after synovectomy or ulnar and radial nerve injury during elbow arthroscopic surgery can sometimes result in sequelae that are difficult to recover from [8]. Various methods have been proposed to solve these problems. Improvements in surgical settings, such as arthroscopic portals and techniques [5], and methods in which images such as computer tomography and magnetic resonance imaging are taken preoperatively to identify the anatomical location, which can then be grasped by the surgeon using navigation [9] and augmented reality technology [10,11], have been reported, but it is difficult to match information obtained from preoperative examinations with the exact location to be treated during arthroscopic surgery. Ultrasonography is thought to be effective for acquiring real-time intra- and extra-articular positional and qualitative information of the tissues and structures. This can also reveal details not visible through the arthroscope view. Methods using superficial ultrasound [12] or ultrasound catheters intraoperatively [13] have been proposed, however, both methods present certain challenges. Superficial ultrasound can be problematic as it is difficult to align the ultrasonography perspective with the view from inside of the joint. Moreover, the intraoperative ultrasound catheter technique uses an intravascular catheter, which is optimized for observation of the vessel wall and not necessarily suitable for evaluation of the intra-articular joint.

The two most used scanning modes in ultrasound systems are amplitude mode (A-mode) and brightness mode (B-mode), which are the basis for other advanced scanning modes such as Doppler mode and motion mode. In A-mode, the reflected ultrasound is displayed in one-dimensional graphical form and can be used to measure the distance between tissues or tissue thickness, while B-mode represents the amplitude peaks seen in A-mode as dots or pixels of varying brightness. Sequential ultrasound pulses can be sent in different directions to form multiple image lines. This process is completed quickly and repeatedly, producing the typical ultrasound image seen on all systems. This mode is used by many ultrasound systems since it provides intuitive positional information about tissue. For example, scanning in B mode is necessary to prevent tissue damage around joints and to determine the physiological state of articular cartilage, and in this study, we developed an intra-articular ultrasonography probe compatible with B-mode scanning that can be used as an examination tool during arthroscopic surgery and conducted a phantom-based proof-of-concept study to verify the observation accuracy of the device.

## 2. Materials and Methods

### 2.1. Setup of an Arthroscopic Ultrasonography Probe System

A schematic diagram of the ultrasonic probe system is shown in Figure 1. A probe rod was specially ordered and made of surgical stainless steel with a diameter of 4.57 mm; the tip was notched in parallel with the axis line, and an ultrasonic element, which transmits and receives ultrasonic waves (ultrasonic vibrator, 12 MHz, 1.5 mm square), was made of a piezoelectric ceramic material and was covered and protected by a polyimide film (Kapton, Du Pont-Toray Co., Ltd., Tokyo, Japan). A 1-mm ferrite rubber packing material was bonded to the back of the transducer to absorb backward free vibration. The acoustic lens made of silicon rubber was bonded to the front of the transducer in the axis line direction on the notched surface to narrow the focus of the beam perpendicular to the scanning cross-section (Figure 1A). This whole probe, including the element and lens themselves, were custom made by Okusonic Co., Ltd. (Saitama, Japan). Ultrasonic sector scanning was performed by rotating the probe around the rotational axis with stepping motor (AZM24AK, Oriental Motor Co., Ltd., Tokyo, Japan) function as a “scanning mechanism,” and its speed was controlled by motor controller (AZD-KD, Oriental Motor Co., Ltd.) (Figure 1C) at its root and rotated coaxially to enable mechanical scanning. A computer-controlled ultrasonic spike pulser receiver DPR-300 (Imaginant, Inc., Pittsford, NY, USA) was used as the ultrasonic transmitter/receiver, and an APX-5040 (Aval Data Co., Ltd., Tokyo, Japan) was used to digitize the received signal.

The signal from the ultrasonic spike pulser was the output from the ultrasonic element of the probe, and the signal reflected by the target tissue was digitized by the APX-5040 digitizer. The amplitude of the signal was converted to intensity on the time axis by luminance modulation in the image processor, and then the information on the position of the ultrasound beam and the time intensity signal were simultaneously plotted in two dimensions by a controlled synchronous scan through the central processing unit to produce a tomographic image of the target tissue (Figure 2).

### 2.2. Evaluation of the System under Acoustic Measuring

The higher the output of the ultrasound probe, the better the signal-to-noise ratio, and the clearer the image can be constructed. However, this also increases tissue temperature and the risk of tissue damage. Before generating and evaluating ultrasound examination images, it was necessary to ensure that the output of the ultrasound probe is within a safe range. The ultrasonic sound field oscillating from the probe was measured using an underwater acoustic intensity measuring device (AIMS, Onda Corporation, Sunnyvale, CA, USA). To measure the intensity of the ultrasonic sound of the central portion, the hydrophone was moved up, down, and away from the probe center so that the sound field could be measured in three dimensions. In this way, AIMS was used to find the point of maximum sound pressure at the center axis of the irradiated surface and to measure the maximum value of negative sound pressure measured at that point. From the viewpoint of safety, the most important biological effects of ultrasound are heating and cavitation. This is due to the fact that the heating effect induces thermal denaturation of proteins, and the cavitation effect leads to cell destruction. Therefore, in order to confirm that the heating and cavitation effects are within the acceptable range, we measured the thermal index (TI) and mechanical index (MI) using the method specified in the IEC (International Electro-technical Commission) 62359:2010 standard and confirmed that they are compatible.

The TI is defined as the ratio of the power used to that required to raise the temperature of the tissue by 1 °C and is calculated using the following equation; $TI=\frac{P/D_{eq}}{C_{TIC}}$, where D_eq_ is the equivalent beam diameter $\left(\frac{4}{\pi }\cdot A_{aprt}\right),$ and A_aprt_ is the sound intensity measured with AIMS. The sound field in the XY plane at 0.3 cm from the irradiated surface was measured and the area corresponding to the −12 dB range from the peak sound pressure value was calculated. C_TIC_ is a coefficient and 40 [mW/cm], according to IEC62359 clause 5.4.2.1; P is the output power [mW] measured with a digital phosphor oscilloscope DTDS3054C (s electronic balance ultrasonic power meter (Tektronix, Inc., Oregon, OR, USA) and the value was less than 2 mW, which is less than the available measurement resolution, and was substituted into the equation as 2 [mW].

$\mathrm{TI}=\frac{2}{40\sqrt{\frac{4}{\pi }\cdot }A_{\mathrm{aprt}}}$ TI was obtained by substituting the above formula.

According to the IEC62359 clause 5.2.2, the MI is calculated using the following formula:$$\mathrm{MI}=\frac{p_{r}}{C_{\mathrm{MI}}\cdot \sqrt{f_{\mathrm{awf}}}}$$
where $C_{MI}$ = 1 [MPa${\mathrm{Hz}}^{-\frac{1}{2}}$], $f_{\mathrm{awf}}\;$is acoustic working center frequency of the ultrasound pulse and was 0.500 [MHz], and $p_{r}$ is the peak rarefaction pressure of the ultrasound wave and was measured using the AIMS by finding the position where the sound pressure was at its maximum at the center axis of the irradiated surface and measuring the maximum value of the negative sound pressure [MPa] at that time. MI is a unitless number that can be used as an index of cavitation bioeffects, and a higher MI value indicates a greater exposure dose and a higher of tissue damage. Levels below 0.23 are generally considered to have no detectable effects even in the most sensitive case of ophthalmic use [14]. MI was obtained by substituting the formula below.
$$\mathrm{MI}=\frac{p_{r}}{\sqrt{0.5}}$$

### 2.3. Verification of the System Using a Phantom

Next, an agar phantom was prepared for observation experiments. As shown in Figure 3, the mold for the phantom was formed with a transparent acrylic board 142 mm in width, 142 mm in length, and 100 mm in height; a lid with holes 7 mm in diameter for passing tissue and 14 holes formed diagonally at intervals of 10 mm was formed vertically. A hole with a diameter of 15 mm was made in the center for inserting the probe, and a cylindrical rod capable of inserting the probe was prepared.

The ulnar, median, and brachial arteries were collected from the cadaveric arm of the Japanese monkey. The nerve and blood vessels were fixed in the hole of the operculum, the tube was connected from the syringe to maintain the bore of the blood vessel, and the vessel was filled with physiological salt solution. The ulnar nerve and brachial artery were fixed longitudinally using nylon threads at a distance of 10 mm from the center, and the median nerve was fixed at a distance of 20 mm.

The agar for medium use (BA-70, Gelation temperature: 35.0 ± 2.0 °C; Ina Food Industry Co., Ltd., Nagano, Japan) was dissolved in the solution and was cooled to 40 °C, so as not to damage the tissue, and then poured in the mold and solidified; finally, the agar hole was filled with physiological saline and the ultrasonic probe was inserted.

Phantom observations confirmed that the nerves had adventitious and funicular structures. The vessel was examined to confirm the lumen structure. Both nerves and vessels were checked for the accuracy of the distance from the probe.

In this experiment, sampling was performed at 14 bits and 100 MHz, which is 10 times the probe’s center frequency of 10 MHz. Synchronous addition was performed 32 times to reduce random noise, and sensitivity time control was set to −60 db at 0.5 cm or less and 6 dB at 2.0 cm or more. Since there is a large difference in signal intensity between hard tissue such as the bone, which strongly reflects ultrasonic waves, and blood vessels, which do not reflect ultrasonic waves well, to selectively amplify signals from blood vessels and nerves, dynamic range processing was performed using the equation below.

Y=85logX, where X is the input value, and Y is the output value. The received data is in a polar coordinate system centered on the needle probe. To convert this data to a Cartesian coordinate system, four points in the polar coordinate system closest to the processing target point in the Cartesian coordinate system were calculated, and the brightness value of the target point was obtained by bilinear interpolation, and two-dimensional ultrasonography images were produced for output.

## 3. Results

### 3.1. Probe Sound Field Property Measurement

The ultrasonic sound field output from the probe was measured using AIMS as shown in Figure 4. Measurements of the ultrasonic sound field output from the probe using an underwater sound intensity measuring device revealed that an A_aprt_ of 9.08 cm^2^ and $p_{r}$ of 0.0107 MPa, resulted in a thermal index (TI) of 0.0147 and a mechanical index (MI) of 0.015. Measurements showed that TI and MI did not exceed 1.0, and the risk related to acoustic output was considered low and acceptable. The intensity distribution was uniform around the probe and within the power guidelines of the American Institute of Ultrasound Medicine (AIUM) and the US Food and Drug Administration (FDA) for ultrasound devices.

### 3.2. Observation of the Agar Phantom

Figure 5A shows an image of a blood vessel. A representative slice of the central vertical axis was then extracted. Low-intensity luminal structures were observed within the high-intensity outer luminal structures. The overall internal structure was of low resolution, although a high contrast was observed due to the water filling the vessel lumen. However, the distance between the probe and the vessel thickness can be accurately determined. Figure 5B shows an image of the ulnar nerve. Representative slices from the longitudinal central region were extracted. As a result of the low resolution, it was difficult to visualize the outer membrane structure, but the funicular structure originating from the inner nerve bundle was barely visible. As with blood vessels, the distance from the probe and the thickness of the nerve can be accurately depicted. The 3D reconstructed images are shown on the right side of each image, and the slices were reconstructed with little two-dimensional displacement due to the vertical movement.

## 4. Discussion

Arthroscopy of the elbow, wrist, and hand continues to advance as a valuable clinical technique in hand surgery that facilitates effective diagnosis and surgery, with its minimally invasive nature as a major advantage. However, as arthroscopy becomes more popular, iatrogenic neurovascular complications have become a problem, with the highest frequency of neurological complications reported in 0.2–3.3% for elbow arthroscopy [7], 0.58% for wrist arthroscopy [15], 1% for carpometacarpal joint arthroscopy (out of the 4% of total complications, only neurological disorders were extracted) [16], and 0.14% for endoscopic carpal tunnel release [17]. These endoscopic procedures in particular are highly technically demanding [2,18], and most complications are thought to be caused by “failure to achieve the procedure” properly [18]. Conventionally, open surgery was performed with a wide surgical field of view, with the surgeon imagining the internal condition of the surgical area by performing medical imaging examinations prior to the surgery. However, in arthroscopic surgery, only small portal incisions are generally made while the anatomical location of important tissues, confirmed by the surgeon using preoperative medical imaging such as MRI, change place according to the limb’s position and swelling. This leads to an inability to visually see important tissues, which is thought to cause the surgical technique errors mentioned above. Another reason is that this surgical technique has become generalized and is increasingly performed by inexperienced surgeons, whereas in the past it was often performed by specialists [19]. To reduce these complications and safely perform arthroscopic surgery, besides an improving in training, ultrasound-guided surgery [20], intravascular ultrasound catheters [21], and near-infrared spectroscopy [22] have been proposed as potential solutions as techniques to visualize important tissues. Ultrasonography is often used to examine the lesions in the upper extremity and has the advantage of visualizing moving structures and has been used for diseases that are manifested by moving joints and surrounding structures in the upper extremity, such as joint locking and dynamic instability, and in surgery for mobile lesions such as ganglions, cysts, and benign tumors. Near-infrared spectroscopy is particularly used to evaluate blood flow in the brain cortex and other tissues with large differences in blood flow changes, such as blood vessels, hemangiomas, and blood flow-rich tumors.

However, in order to combine these technologies with arthroscopic surgery and improve surgical safety, there are limitations such as difficulty in securing working space, difficulty in aligning ultrasound images with endoscopic images, and the need for further knowledge about the consistency with anatomy and pathology, which have led to a demand for an ultrasound probe with multiple frequencies that can be used with the same ease of use as an arthroscopic device [13].

The ultrasound arthroscope device tested in this study offers a promising solution to this problem. It provides intraoperative dynamic information that cannot be obtained with preoperative imaging such as CT or MRI, allowing the surgeon to see anatomical and qualitative tissue changes according to the surgical situation and to perform more accurate surgery. Therefore, the probe is rigid and can be easily manipulated intraoperatively in the same way as conventional endoscopic probe. The use of sector scanning, rather than circular scanning, and a good pulse shape from the transducer helped to maintain a decent spatial resolution, although increasing the density of the transducer elements will likely be necessary to improve the resolution and tissue sorting ability. For example, in arthroscopic ganglionectomy, the ability to perform ultrasonography from within the joint in cases where the stalk cannot be identified by arthroscopy will allow the surgeon to know exactly where in the joint capsule needs to be resected and will also reduce the risk of injury to the radial artery or median nerve. This will also prevent nerve damage and brachial artery damage due to technical reasons in elbow arthroscopic surgery and enable evaluation of the biological activity of articular cartilage and ligament tissue, for example, in the evaluation of free bone fragments in osteochondritis dissecans, leading to more accurate surgery. Furthermore, considering the challenges of securing work space for arthroscopy and the limitations of aligning ultrasound and endoscopic images, it is technically possible to enhance surgical efficiency and safety by using arthroscopic augmented reality (AR) technology [11,23] to superimpose ultrasound findings on endoscopic camera images and to simultaneously visualize conditions outside the arthroscope field of view.

Overall, owing to several mechanical and technical limitations, this study could only provide a proof-of-concept validation using an agar phantom. However, this study made it possible to recognize potential complications and limitations in the spatial discrimination capability of current technology. Based on the findings from this study, we are charting a future course that includes a dual focus. The next phase of ultrasonic arthroscopic device development should involve enhancing resolution and tissue sorting by increasing the number of ultrasonic elements. Additionally, the design of the current probes, while suitable for elbow joints, is too large for wrist joints and must be slimmed down. This redesign will require careful planning to add more elements without exceeding space limitations. Simultaneously, we are also preparing for a first-in-human study, marking another significant step in our research trajectory. In the future, ultrasound arthroscopy may help safely perform more difficult procedures in endoscopic surgeries (although the density of elements needs to be increased to improve the resolution for better tissue sorting).

## 5. Patents

Nobuo Niimi, Masato Mori, Hitoshi Hirata, Shintaro Oyama: The joint surgery support devices, ultrasonic probes, and joint surgery systems JP-P2021-186308A represent a patent resulting from the work reported in this manuscript.

## Figures and Tables

**Figure 1 jcm-12-05727-f001:**
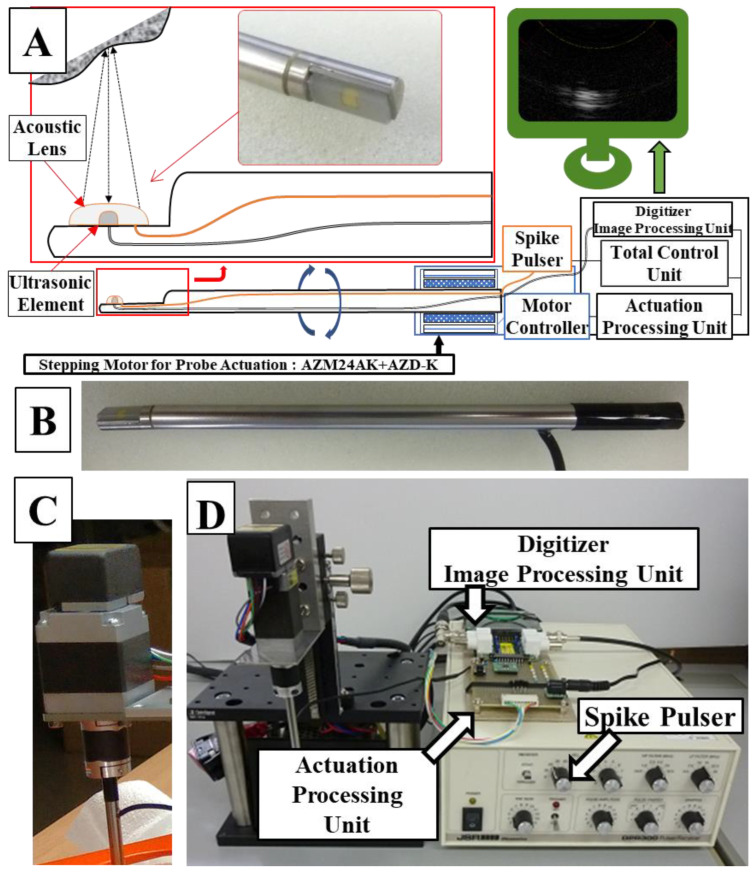
A system schema (**A**) of the manufactured probe (orange line: acoustic pulse line, black line: signal line from the ultrasonic element to the digitizer), a photograph (**B**) of the actual probe, a stepping motor (**C**) for controlling rotation of the probe, and an overall mechanism diagram (**D**).

**Figure 2 jcm-12-05727-f002:**
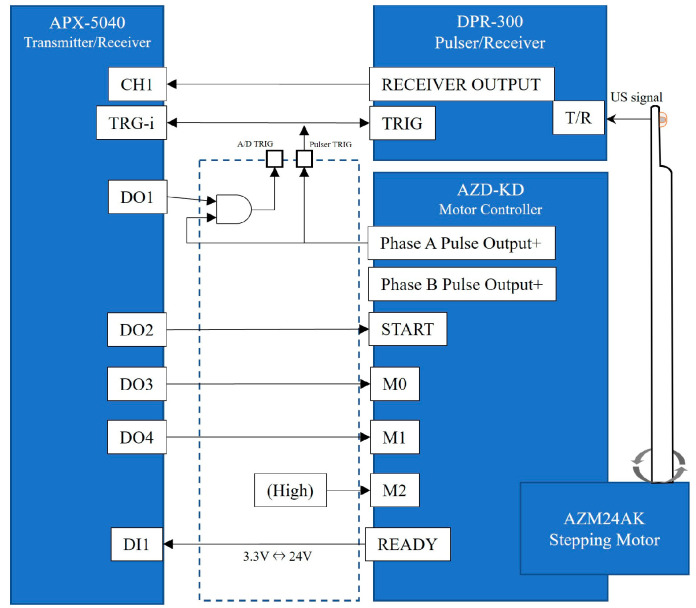
A detailed system block diagram. The AZD-KD motor controller can execute motions with multiple motion modes, origins, and speeds set in advance by inputting signals to M0, M1, and M2. M0 and M1 were controlled from the APX-5040 digital output channel, D03 and D04, and M2 were fixed at a high level. START/READY on AZD-KD are a signal to notify the start/completion of motion. In this experiment, the registered motion settings of AZD-KD were set to reciprocate between angles −45° and 45° at a speed of 5000 r/min.

**Figure 3 jcm-12-05727-f003:**
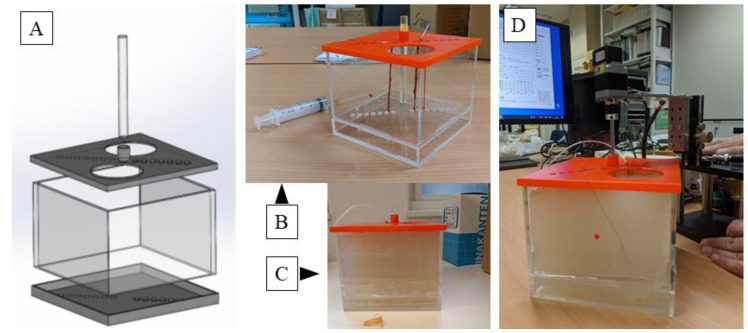
A computer aided design (CAD) model of the mold for the agar phantom. Lids have a width of 142 mm, a length of 142 mm, and a height of 100 mm, each having a diameter of 7 mm and having 14 holes each for passing tissue on a diagonal line at intervals of 10 mm on the lid. A hole of 15 mm diameter is made in the center of the top lid for inserting the probe (**A**); an actual molded plastic mold. Nerves and blood vessels can be fixed here. A tube is connected from a syringe to maintain the lumen of the blood vessel, and the vessel is filled with physiological saline (**B**); and the agar dissolved therein is cooled to 40 °C and poured and solidified (**C**); and finally physiological saline is put into the central hole, and an ultrasonic probe is inserted and observed (**D**). Please find the Appendix A.

**Figure 4 jcm-12-05727-f004:**
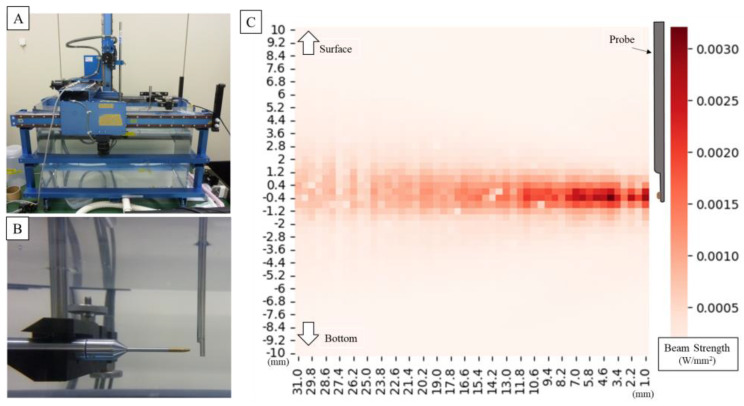
Setting of underwater acoustic intensity measuring device (**A**). It is measured in pure water at room temperature. The hydrophone moves as it measures around the probe (**B**). The image below shows the measured amplitude overlaid on a photo. Graph plotting ultrasonic intensity around the probe (for the first measurement) (**C**); the waveform remains almost the same for all three measurements. The intensity distribution is uniform around the probe, and the strongest point shows sufficient intensity for an ultrasonic measurement probe.

**Figure 5 jcm-12-05727-f005:**
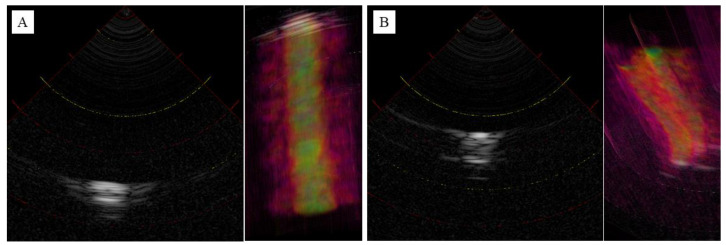
Ultrasonography images of blood vessel (**A**) and ulnar nerve (**B**) in 2D and 3D reconstruction. Intensity represented in grayscale (high intensity: white, low intensity: black) in 2D, heatmap (high intensity: yellow, low intensity: red) in 3D.

## Data Availability

The data that support the findings of this study are available at Nippon Sigmax. Restrictions apply to the availability of these data, which were used under license for this study. Data are available on request from the corresponding author, S.O. with the permission of Nippon Sigmax.

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
