# Peer review of "Intra-Articular Ultrasonography Probe for Minimally Invasive Upper Extremity Arthroscopic Surgery: A Phantom Study"

_jcm, 2023, doi:10.3390/jcm12175727_

Round 1
Reviewer 1 Report
First of all congratulations for your article. You can find my comments and suggestions below.
1- It would be helpful if the references in the introduction section were more up-to-date and more numerous.
2-It would be useful to increase the visual content and number used in the article.
Author Response
Dear Reviewer.1
Thank you for the kind review.
We have updated introduction, references, and visual content.
(https://youtube.com/shorts/sdg-uMW3wVo)
We would appreciate your confirmation.
Reviewer 2 Report
This study is well written but has major flaws.
This is an experimental study.
How does the study translate to real life? Is this study relevant in real life?
The authors should at least have a case series before they publish this paper.
The scientific relevance is low. Goo idea but will not be able to translate in real life.
References are good.
I do not detect plagiarism
English language is okay.
Author Response
Dear Reviewer 2,
Thank you for your kind review.
We have followed your advice and modified some content.
We have described in detail how this technology can be used in actual clinical practice.
As for how this will help in actual surgery, as noted in the discussion, for example, in arthroscopic ganglionectomy, ultrasound can be performed from within the joint when the stem cannot be seen by arthroscopy, allowing the surgeon to know exactly where in the joint capsule needs to be resected, and also reducing the risk of injury to the radial artery or We believe that it also reduces the risk of injury to the median nerve. I have revised the text of the discussion to discuss the advantages.
Regarding the use of this technology in the cases you mentioned, we are planning to conduct a clinical study with this paper as proof of the technical background, and we would like to conduct a first-in-human study as you mentioned in the near future.
Reviewer 3 Report
On your abstract
The title is ok
- It is unclear what you mean by "it provided sufficient output as an ultrasound measurement probe." Could this be clarified or expanded upon?
- term "A-mode imaging" might be unfamiliar to a broader audience. Consider explaining
On your Introduction
- "the technical requirements for arthroscopic surgery of the upper limb joints are high, and not many surgeons are able to perform the surgery yet" could be clarified. Do you mean that not many surgeons have the necessary training or that the procedure is not widely adopted?
- Avoid repetitive phrases such as "upper extremity arthroscopic surgeries" and "arthroscopic surgery of the upper limb joints" - choose one term and stick with it throughout the introduction to maintain consistency
- The sentence starting with "Ultrasonography is thought to be suitable..." is too long and complex. Try breaking it into multiple sentences.
On your Methods
- "MI is a unitless number that can be used as an index of cavitation bioeffects, and a higher MI value indicates greater exposure." What do you mean by "exposure" here? Do you mean "risk" or "likelihood"?
- Figure 1 - the writing and text is too small and cannot be read. Please increase the size of this figure.
On your Results
They are good and nicely written
On your Discussions
- The term "commoditized" might not be the best choice here, as it typically refers to products or services that have become so common they are interchangeable.
- Considering the limitations in securing working space and aligning ultrasound images with endoscopic images, how might technological advancements aid in integrating ultrasound-guided surgery with arthroscopic techniques to improve surgical safety?
- Given the initial success of the ultrasound arthroscope device tested in this study, what are the next steps in its development to further improve resolution and tissue sorting ability, and how might these improvements enhance the outcomes of complex endoscopic surgeries?
Author Response
Dear Reviewer 3,
Thank you for your kind review.
We have followed your advice and modified some content.
In response to some of your comments, we have revised the manuscript as follows;
- It is unclear what you mean by "it provided sufficient output as an ultrasound measurement probe." Could this be clarified or expanded upon?
=> We have revised the description you pointed out to a more clear wording, "while confirming that sufficient output level performance can be obtained as an ultrasonic measurement probe."
- term "A-mode imaging" might be unfamiliar to a broader audience. Consider explaining
We have added description about A and B-modes.
- "the technical requirements for arthroscopic surgery of the upper limb joints are high, and not many surgeons are able to perform the surgery yet" could be clarified. Do you mean that not many surgeons have the necessary training or that the procedure is not widely adopted?
- Avoid repetitive phrases such as "upper extremity arthroscopic surgeries" and "arthroscopic surgery of the upper limb joints" - choose one term and stick with it throughout the introduction to maintain consistency
=>We have revised as “the technical requirements for upper extremity arthroscopic surgeries are high, and not many surgeons are yet trained enough to perform the surgery.”
- The sentence starting with "Ultrasonography is thought to be suitable..." is too long and complex. Try breaking it into multiple sentences.
=>We have corrected this as you pointed out.
- "MI is a unitless number that can be used as an index of cavitation bioeffects, and a higher MI value indicates greater exposure." What do you mean by "exposure" here? Do you mean "risk" or "likelihood"?
=>We have revised as “MI is a unitless number that can be used as an index of cavitation bioeffects, and a higher MI value indicates greater exposure dose and the higher the risk of tissue damage.”
- Figure 1 - the writing and text is too small and cannot be read. Please increase the size of this figure.
=>We have rearranged the Fig.1.
- The term "commoditized" might not be the best choice here, as it typically refers to products or services that have become so common they are interchangeable.
=>We have revised as “surgical technique has become generalized”.
- Considering the limitations in securing working space and aligning ultrasound images with endoscopic images, how might technological advancements aid in integrating ultrasound-guided surgery with arthroscopic techniques to improve surgical safety?
=>considering the challenges of securing work space for arthroscopy and the limitations of aligning ultrasound and endoscopic images, it is technically possible to enhance surgical efficiency and safety by using arthroscopic augmented reality (AR) technology to superimpose ultrasound findings on endoscopic camera images and to simultaneously visualize conditions outside the arthroscope field of view. (we also revised the manuscript)
- Given the initial success of the ultrasound arthroscope device tested in this study, what are the next steps in its development to further improve resolution and tissue sorting ability, andhow might these improvements enhance the outcomes of complex endoscopic surgeries?
=>Based on the findings from this study, we are charting a future course that includes a dual focus. The next phase of ultrasonic arthroscopic device development should involve enhancing resolution and tissue sorting by increasing the number of ultrasonic elements. Additionally, the design of the current probes, while suitable for elbow joints, is too large for wrist joints and must be slimmed down. This redesign will require careful planning to add more elements without exceeding space limitations. (we also revised the manuscript)